# Programmed magnetic manipulation of vesicles into spatially coded prototissue architectures arrays

Qingchuan Li[1,2], Shubin Li[1,2], Xiangxiang Zhang[1], Weili Xu[1] & Xiaojun Han[1]*

In nature, cells self-assemble into spatially coded tissular configurations to execute higher-order biological functions as a collective. This mechanism has stimulated the recent trend in synthetic biology to construct tissue-like assemblies from protocell entities, with the aim to understand the evolution mechanism of multicellular mechanisms, create smart materials or devices, and engineer tissue-like biomedical implant. However, the formation of spatially coded and communicating micro-architectures from large quantity of protocell entities, especially for lipid vesicle-based systems that mostly resemble cells, is still challenging. Herein, we magnetically assemble giant unilamellar vesicles (GUVs) or cells into various microstructures with spatially coded configurations and spatialized cascade biochemical reactions using a stainless steel mesh. GUVs in these tissue-like aggregates exhibit uncustomary osmotic stability that cannot be achieved by individual GUVs suspensions. This work provides a versatile and cost-effective strategy to form robust tissue-mimics and indicates a possible superiority of protocell colonies to individual protocells.

[1] State Key Laboratory of Urban Water Resource and Environment, School of Chemistry and Chemical Engineering, Harbin Institute of Technology, 92 West Da-Zhi Street, Harbin 150001, China. [2] These authors contributed equally: Qingchuan Li, Shubin Li *email: hanxiaojun@hit.edu.cn

D uring the evolution of life, one of the major transitions is the appearance of multicellular systems with spatially coded cell types[1,2], which communicate and cooperate to exhibit higher-order collective behaviors in the form of tissues or organs. Mimicking these systems via the controlled assembly of synthetic cell-like entities is expected to result in important implications for the fabrication of artificial living systems and promising applications in the field of tissue engineering[3]. So far, various kinds of protocell entities, such as liposomes[4–6], polymersomes[7], proteinosomes[8], and water-in-oil emulsion droplets[9–11] have been integrated into rudimentary tissue-like assemblies that exhibit higher-order behaviors as a collective including communication, deformation, signaling, and differentiation. However, except the series of breakthrough studies based on water-in-oil emulsion networks, most of the current tissue-like assemblies are amorphous aggregates of some protocell entities, especially for the lipid vesicle-based systems that most closely resemble cells[12]. In a recent breakthrough, Ces and coworkers[13] sculpted small group of vesicles into defined spatial organization using optical tweezers. The assembly of large quantity of vesicle-based protocell entities into spatially coded and communicating micro-architectures to mimic the existence form of natural tissues remains a considerable challenge.

As a versatile, noninvasive, and cost-effective strategy, magnetic manipulation has been increasingly exploited for the scalable assembly of magnetic and nonmagnetic objects into two-dimensional (2D) or three-dimensional (3D) metastructures based on their responses to inhomogeneous magnetic field in two mechanisms: positive magnetophoresis (moving to areas with maximized field intensity) for magnetic objects[14,15], and negative magnetophoresis (moving to areas with minimized field intensity) for objects with lower magnetic susceptibility than that of suspension media[16,17]. Compared with the well-established study and wide application of the first mechanism, the investigation of the second mechanism is still in its infancy, but attracts intense attentions in recent years because of its universal applicability for different kinds of inanimate and living materials[18,19], and unique manipulation behaviors for the objects going beyond Earnshaw's theorem[20]. Early study in this area often required quite high magnetic field intensity. However, a recent revolution by introducing paramagnetic dispersing environment brought up a magneto-Archimedes effect[21], enabling the manipulation of nonmagnetic objects under weak magnetic field. Based on this effect, 2D colloidal particles lattice was assembled on Ni grid with different morphologies in paramagnetic $Ho(NO_3)_3$ solution[22]. Simple 3D structures, for example, the spheroidal tissue-like models, have also been obtained via magnetic levitation devices in a more biocompatible gadolinium-based nonionic paramagnetic solution[23]. However, the application of this technique for the formation of more complicated 3D aggregates, for instance, the spatially coded tissue-like giant unilamellar vesicles (GUVs) assemblies, has rarely been reported.

Herein, we describe the scalable magnetic assembly of cell-mimic (GUVs) colonies with tissue-like complex 3D organizations using a stainless steel (SS) mesh with patterned microwells in a paramagnetic solution media. The independent and collaborative influences of the microwell parameters (morphology and arrangement) and directions of external magnetic field on GUVs colonies formation with different spatial organizations are investigated. Cascade enzyme reactions among these spatially organized structures are engineered. Our work provides a method to form higher-order tissue-like structures for synthetic biology, tissue engineering, and the study of spatially compartmented chemical reactions, and exhibits a further step for the controlled magnetic manipulation.

## Results

**The setup for GUVs assembly**. The assembly of diamagnetic GUVs was carried out on a SS mesh (thickness $\approx 100\,\mu m$) with patterned microwells (Supplementary Fig. 1) by mixing GUVs mother dispersion electroformed in 400 mM sucrose solution (Supplementary Fig. 2) with isotonic paramagnetic $MnCl_2$ or Gadobutrol solution. The inner volume of GUVs was 400 mM sucrose solution, and the outside solution was an isotonic mixture of sucrose and paramagnetic compounds with relative lower density. The GUVs encapsulated with sucrose solution were heavier than their surroundings, so when no magnetic field was applied, GUVs homogeneously deposited on the microwells and grids under gravity (Supplementary Fig. 3). After the magnetic field was applied from the bottom, the SS mesh exhibited a paramagnetic response, i.e., magnetic moments parallel to the external magnetic field were generated in the SS mesh (Supplementary Fig. 4a), which resulted in the formation of magnetic field gradient microenvironments around the SS mesh because of the interplay between magnetized magnetic field from SS mesh and external magnetic field from the magnets. These microenvironments drove the aggregation of GUVs that deposited around the SS mesh under gravity to form tissue-like colonies in the paramagnetic media containing $MnCl_2$. For a GUV in magnetic field with radius of $R$ at position $\mathbf{r}$, the magnetostatic potential energy $U(\mathbf{r})$ was given by[22]

$$U(r) = -2\pi R^3 \mu_0 \frac{\chi_G - \chi_S}{\chi_G + 2\chi_S + 3} |\mathbf{H}(\mathbf{r})|^2, \qquad (1)$$

where $\mu_0$ is the magnetic permeability of vacuum, $\chi_G$ and $\chi_S$ are the magnetic susceptibilities of GUVs and paramagnetic solution, respectively, and $\mathbf{H}(\mathbf{r})$ is the magnetic field at the position of the GUVs. For diamagnetic GUVs ($\chi_G < \chi_S$), the potential energy $U(\mathbf{r})$ was strictly positive as implied by Eq. (1), so GUVs tended to move towards regions with minimum magnetic field in the gradient microenvironments for lower $U(\mathbf{r})$. Therefore, it can be expected that the spatial organization of GUVs colonies can be determined by the distribution of magnetic field around the microwells. In this work, magnetic fields with three directions versus the SS mesh surface, i.e., vertical (Fig. 1a), horizontal (Fig. 2a), and inclined (Fig. 2d) magnetic fields, were adjusted to modulate the magnetic field microenvironment around the SS mesh for the assembly of GUVs in paramagnetic media. The assembly device contained a cover slip, the SS mesh clinging to the cover slip, and a Teflon cell enclosing the SS mesh (Supplementary Fig. 5).

**GUVs assembly under vertical magnetic field**. The vertical external magnetic field versus the SS mesh surface was provided by putting the device on the top center of a permanent magnet (Fig. 1a). As indicated by the horizontal (top in Fig. 1b) and vertical (bottom in Fig. 1b) central section of the simulated magnetic field distribution across the microwells, local microenvironments with lower magnetic field strength than surrounding space were generated in the patterned microwells. So diamagnetic GUVs tended to aggregate in the microwells under gravity and negative magnetophoresis. Moreover, the strength of the magnetic field in the microwells is also inhomogeneous, radially decreased from the center to the well walls, which caused the preferential localization of GUVs around the microwells to form toroidal microstructures. This phenomenon is more superficially similar to the traditional manipulation behavior for magnetic objects rather than diamagnetic entities. In previous studies related to magnetic levitation or colloidal assembly at 2D

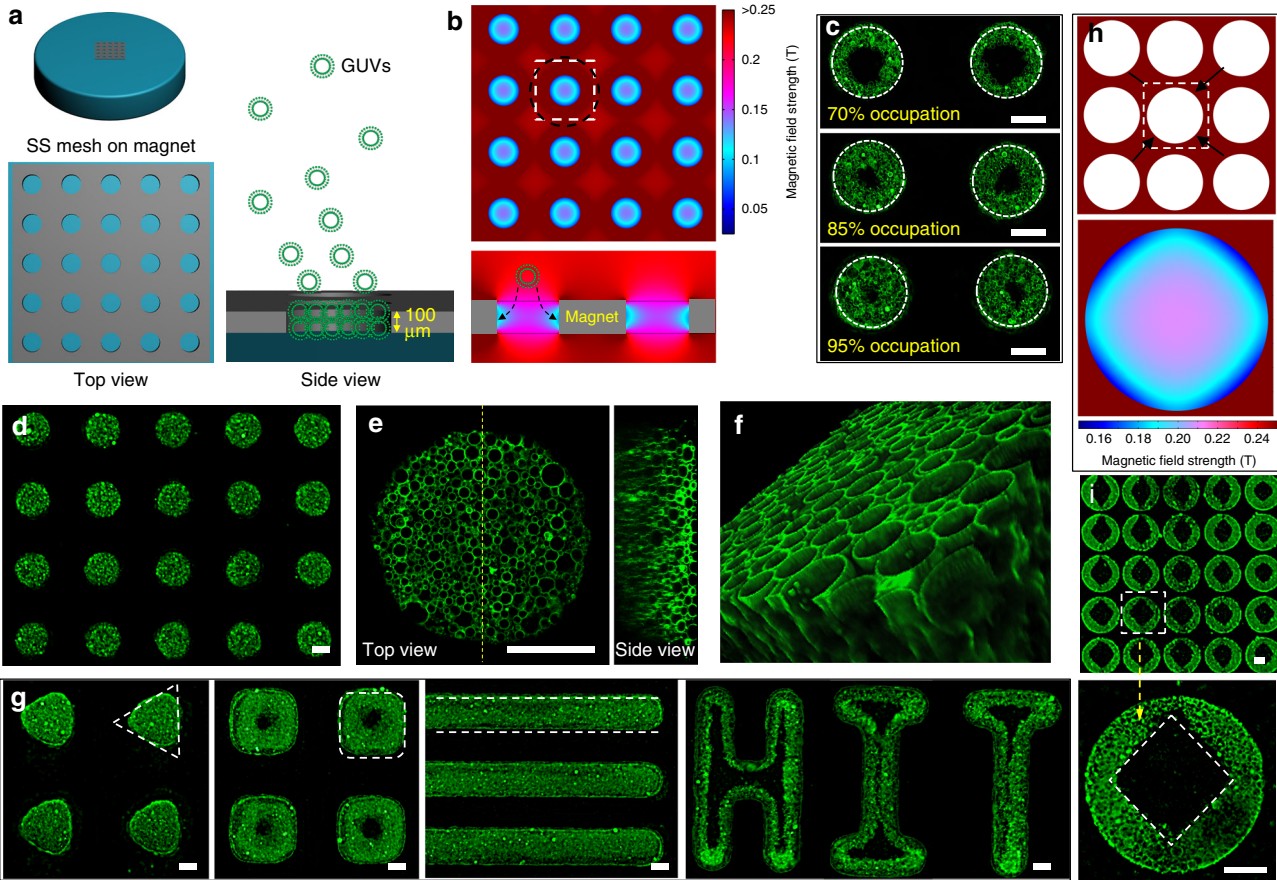

**Fig. 1 Assembly of giant unilamellar vesicles (GUVs) on stainless steel (SS) mesh under vertical magnetic field. a** Schematic illustration of the device for GUVs assembly: a SS mesh placed on the top center of a magnet. **b** Horizontal (top) and vertical (bottom) central section of the simulated magnetic field distribution across the microwells. The white dash box and the black dash circle respectively indicated the unit of the microwell array and the approximating unit for decentralized microwells. The black dash arrow indicated the preferential localization of GUVs around microwell wall.
**c** Fluorescence images of the GUVs colonies with different extent of occupation of the microwells. The white dash circle indicated microwell wall.
**d** Fluorescence image of GUV colony arrays formed in SS mesh with microwell diameter of 250 μm. **e** Top view and side view along the yellow dash section line of the GUVs colony taken by a laser confocal microscope. **f** A 3D image of GUVs colony obtained from serial sections of images in the Z-stacks taken by a laser confocal microscope. **g** Fluorescence images of GUVs colonies with different morphologies: from left to right, triangular, square, striped, and HIT-like assemblies. The dash triangle, rectangle, and line illustrated rough outline of GUVs colonies. **h** The schematic and simulated magnetic field distribution of SS mesh with densely packed microwells. The white dash box presented the unit of the microwell array. The black arrows indicated the corners.
**i** Fluorescence images of the Chinese ancient coin-like round GUVs colonies with square holes formed using the SS mesh with densely packed microwells. The bottom is the enlarged image in the dash box of top image as indicated by the yellow dash arrow. The white dash box in bottom image indicated the square hole in GUVs colony. The scale bars are 100 μm.

surfaces, diamagnetic objects tended to run away from the magnets or the magnetized Ni grids, while magnetic materials tended to be attracted and collected by them[22]. The interplay of the magnetic field from the magnet and the magnetized SS mesh resulted in the paramagnetic manipulation-like collection of GUVs on the microwell walls. With the increase of time and amount of GUVs, the microwells were gradually occupied by GUVs from the well walls to the center (Fig. 1c and Supplementary Fig. 6). Complete occupation of the microwells to form columnar GUVs colonies arrays (Fig. 1d) was realized with incubation time of 2 h in 0.04 mg mL$^{-1}$ GUVs solution (volume $= 300$ μL, $V_{\text{GUVs mother solution}}/V_{\text{MnCl}_2 \text{ solution}} = 1/4$). The top view and side view from the laser confocal microscope indicated the close packing of GUVs in the colony (Fig. 1e). The columnar 3D structure of the tissue-like aggregates can be recognized from a 3D image constructed from serial sections of images in Z-stacks (Fig. 1f). There existed the adhesive van der Waals force and different kinds of repulsive forces, including undulation, hydration, and electrostatic forces, among GUVs. The balance of these

forces determined whether the GUVs were repulsive or adhesive. However, this had no influence on the close packing of GUVs. The adhesive GUVs in assembly solution containing MnCl$_2$, and the repulsive GUVs in 400 mM nonionic Gadobutrol solution can both form closely packed GUVs colonies (Supplementary Fig. 7). The driving forces for GUVs assembly and close packing were the magnetic force and the gravity. The gravity facilitated the deposition of GUVs around the mesh and the magnetic force promoted their aggregation at region with lower magnetic field rather than deform them (Supplementary Fig. 8).

As shown by Fig. 1e and Supplementary Fig. 9a, the electroformed polydisperse GUVs displayed heterogeneous size distribution in the microwells. From the area close to the microwell wall to that close to the center, the average diameter of GUVs gradually decreased. This phenomenon was more pronounced for the GUVs colony that partially occupied the microwells via using fewer GUVs (Supplementary Fig. 9b). This is because larger GUVs settled more quickly under gravity than smaller ones[24]. Based on this size-dependent assembly

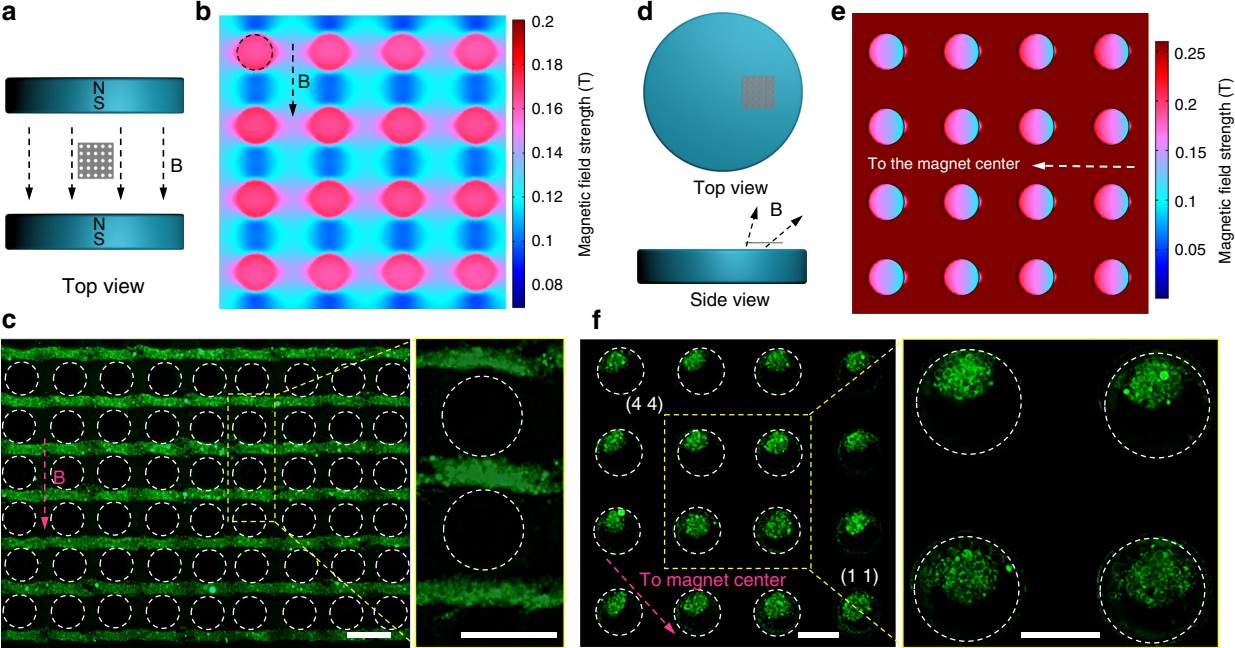

**Fig. 2 Assembly of GUVs on the SS mesh under horizontal and inclined magnetic field. a** The schematic of the device for horizontal magnetic field. **b** Simulated magnetic field distribution on the top surface of the SS mesh under horizontal magnetic field. **c** Fluorescence images of the GUVs colonies formed under horizontal magnetic field. The right image is the enlarged image in the yellow dash box of the left image. **d** The schematic of the device for inclined magnetic field (with directions between the vertical and horizontal magnetic fields) by putting the SS mesh on one side of the top of the magnet. **e** Simulated magnetic field distribution at the bottom surface of the SS mesh under inclined magnetic field. **f** Fluorescence images of the GUVs colonies under inclined magnetic field. The right image is the enlarged image in the yellow dash box of the left image. The dash circles in **b**, **c**, **e**, and **f** indicate the microwells. The scale bars in **c** and **f** are 200 μm.

phenomenon, we programmed the formation of GUVs colony with alternating GUVs layers of different average sizes (Supplementary Fig. 9c) via the successive adding of GUVs for two times.

The GUVs in the colony were mutually isolated and retained the integrity of their compartmentalized interior. No fusion/ hemi-fusion events and leakage of compartmentalized fluorescent molecules were observed in the experiments (Supplementary Movie 1 and Supplementary Figs. 10, 11). However, these GUVs displayed no individual transitional motion (Supplementary Movie 1 and Supplementary Fig. 10), but behaved as a jammed and consolidated aggregate with collective stability due to their close packing. When magnetic field was removed and the SS mesh was inverted to provide a harmful gravity field that promoted the GUVs colony disassembly, the contact forces[25–27] among the closely packed GUVs kept the GUVs colony stable, regardless of whether the GUVs were mutually repulsive or adhesive (Supplementary Fig. 7).

The morphology of the obtained GUVs colonies was directly related to the shape parameters of the microwells. The size of the columnar GUVs colonies can be modulated by the diameter of microwell templates (Supplementary Fig. 12). Columnar GUVs colonies with aspect ratios (height/diameter) of 0.5 (Fig. 1d), 1.0, and 1.5 (Supplementary Fig. 13) can be obtained via the variation of the microwell aspect ratios. Moreover, through varying the design of the structure of the microwells, GUVs colonies with different morphologies, including triangular, square, striped, and HIT-shape assemblies, were obtained (Fig. 1g).

From abovementioned results, we have demonstrated that the magnetic field distribution in the microwells was the result of the interplay of the magnetized magnetic field from SS mesh and the external magnetic field from the magnets. In microwells, the magnetized magnetic field was in opposite direction to the external magnetic field, which weakened the magnetic field

strength. Therefore, the magnetic field distribution inside each well can be adjusted by the SS mesh layout. For SS mesh with decentralized microwells (Fig. 1b), the square unit of the microwell arrays (white dash box) can be approximately considered as a circular region (black dash circle). The magnetic field was radially decreased from the center to the well walls (Fig. 1b), resulting in axisymmetric columnar GUVs colonies (Fig. 1c, d). However, when the microwells were quite hugging (Fig. 1h), the square corners (indicated by the black arrows) generated more magnetized magnetic field, which resulted in the weakening of magnetic field strength in greater degree at the part of microwells adjacent to the corner (bottom image in Fig. 1h). The magnetic field microenvironments in the microwells then guided the assembly of GUVs into Chinese ancient coin-like round GUVs colonies with square holes (Fig. 1i). GUV colonies with other morphologies, such as round colonies with oval, elliptical, heart-shaped, half-round, or hexagonal holes, and striped colonies with waved edges, can be predicted to generate by varying the spatial organization of the microwells according to the simulated results (Supplementary Fig. 14). Taken together, from abovementioned results, GUVs colonies with various morphologies were formed by the variation of microwell morphologies and spatial organization under vertical magnetic field.

**GUVs assembly under horizontal and inclined magnetic field.** The above text addressed the influence of the morphology and spatial organization of microwells on GUVs colonies formation under external vertical magnetic field. The following text will discuss the influences of external magnetic field with other directions, i.e., horizontal magnetic field and inclined magnetic field, on GUVs colonies formation. Under horizontal external

magnetic field by locating the SS mesh between two face-to-face magnets (Fig. 2a), the region with minimum magnetic field strength located on the frame of the SS mesh (Fig. 2b). So stripped GUVs colonies perpendicular to external magnetic field were formed on the frames rather than inside the microwells (white dash circles) (Fig. 2c). Inclined external magnetic field containing both horizontal and vertical components was provided by putting the SS mesh on one side of magnet top surface (Fig. 2d), which resulted in Janus distribution of magnetic field strength at the bottom surface of the microwells. The part of individual microwell away from the magnet center was in low magnetic field strength (blue), while the opposite part was in high magnetic field strength (red) (Fig. 2e). Therefore, as presented by the fluorescence images, GUVs aggregated at the part of individual microwell away from the magnet center rather than the other side (Fig. 2f). Moreover, a gradient occupancy of the microwells by GUVs was observed in the experiments (Fig. 2f and Supplementary Fig. 15). From the microwells near the magnet center to those close to the magnet edge, the amount of GUVs in microwells gradually decreased. For example, more than half area of microwell (1 1) in Fig. 2f was filled by GUVs, while microwell (4 4) away from the magnet center was only occupied for about one third of the area. We ascribe the gradient location of GUVs to the non-uniform spatial distribution of the inclined magnetic field. As presented by the simulated magnetic field distribution around the magnet (Supplementary Fig. 16), the ratio of horizontal component to the vertical component increased from the magnet center to the edge, which resulted in decreased areas with low magnet flux density in Fig. 2e for GUVs to occupy.

**Formation of hybrid GUVs colonies**. Biological tissues are composed of coded microstructures with different cell types. To mimic this structural complexity, we magnetically manipulated the assembly of two kinds of GUVs, i.e., 1,2-dioleoyl-$sn$-glycero-3-phosphoethanolamine-N-(7-nitro-2-1,3-benzoxadiazol-4-yl) (NBD PE) labeled GUVs with green fluorescence (gGUVs) and 1,2-dihexadecanoyl-$sn$-glycero-3-phosphoethanolamine with triethylammonium salt (TR DHPE) labeled GUVs with red fluorescence (rGUVs) into configurations with different spatial organizations. Parallel coding with fully mixed gGUVs and rGUVs was obtained for the assembly event of pre-mixed GUVs under vertical magnetic field (Fig. 3a, Supplementary Fig. 17). However, higher-order structures with serially coded GUVs colonies can be obtained via the alternate addition of different GUVs or the application of different magnetic fields. Firstly, patterned layer-by-layer gGUVs and rGUVs colonies were observed via the alternate addition of these two kinds of GUVs under vertical magnetic field (Fig. 3b, Supplementary Figs. 18 and 19). The ratio of these two GUVs colonies in the microwells can be modulated by varying the added amount of different GUVs (Supplementary Fig. 20) and the structure of the microwells sculpted morphologies of the microarchitecture (Supplementary Fig. 21). The top and side view of the microstructures under laser confocal microscope confirmed their coaxially coded configuration (Supplementary Fig. 22). Secondly, asymmetrically configured microstructures with two different GUVs colonies were obtained via the alternate application of two inclined magnetic fields (Case 1, Fig. 3c and Supplementary Fig. 23) or one inclined one and another vertical one (Case 2, Fig. 3d and Supplementary Fig. 23) for the chronological trapping of two GUV types. In case 1, the two inclined magnetic fields had different directions, with one of them provided by putting the SS mesh on one side of the top surface of the magnet and the other one provided by putting it at the opposite side. In case 2, except for the rGUVs colonies

that were located opposite to the pre-trapped gGUVs in the microwells, an additional rGUVs layer was formed near the pre-trapped GUVs layer as indicated by the white arrows. This can be attributed to the shaping effect of the pre-trapped GUVs colonies on the spatial distribution of magnetic field in the microwells, resulting in local weak magnetic field around them for GUVs aggregation (Supplementary Fig. 4b). Thirdly, grid-like aggregates composed of orthogonal gGUVs and rGUVs stripes (Fig. 3e, Supplementary Fig. 24) can be generated by the successive application of two horizontal magnetic field perpendicular to each other for the respectively trapping of gGUVs and rGUVs. Finally, through the successive application of one vertical magnetic field for gGUVs and two perpendicular horizontal magnetic fields for rGUVs, we obtained more complex structures containing columnar gGUVs colonies in the microwells and meshed rGUVs on the grid (Fig. 3f, Supplementary Fig. 24).

**Versatility of the magnetic assembly method**. Taken together, via the modulation of SS mesh parameters (microwell morphologies and organizations) and external magnetic field, we proposed a robust and cost-effective method to manipulate large quantity of GUVs into tissue-mimic microstructures with different morphologies and spatial organizations. Moreover, this method can also be applied for the assembly of GUVs on Ni wires to form 1D colonies (Supplementary Fig. 25), on Ni foam to form 3D networks (Supplementary Fig. 26), and even along the scratch on a stainless sheet to generate 1D colonies arrays (Supplementary Fig. 27). Other materials, including gel particles, emulsions, bubbles and even cells, can all be trapped around the support to form defined structures (Supplementary Fig. 28). The versatility, universality, simplicity, and scalability of this method endowed it with potential boom of applications in synthetic biology, tissue engineering, photic and electronic devices fabrication, the engineering of electrode surfaces, and the quality control of iron materials, etc. In this paper, we mainly focus on their potential as proto-tissues to provide a stable environment for individual cell-mimic GUVs, and to spatialize biochemical reactions.

**Osmotic stability of the GUVs colony**. As cell mimics, individual GUVs suspensions are very fragile. For example, an imbalanced osmotic (hypotonic or hypertonic) condition can easily cause their deformation or rupture[28–30]. This impeded their application in advanced synthetic cells development, cell biology, and biosensing, etc. However, in this work, when GUVs were magnetically gathered to form GUVs colonies, they displayed uncustomary stability. For colonies composed of GUVs encapsulating 400 mM sucrose, no morphology change for individual GUVs was observed in a hypotonic condition provided by pure water, and hypertonic conditions provided by 1000 mM glucose, 500 mM NaCl and even 333 mM $CaCl_2$ (Supplementary Fig. 29). The osmotic stability of the GUVs colony originated from the resistance of the closely packed GUVs in the colony as a collective to the external osmotic shock. In hypotonic condition provided by pure water, the reinforced mechanical stability of individual GUVs from the crowded GUVs surroundings impeded their rupture and maintained their morphologies. In hypertonic conditions, GUVs in unbalanced osmotic condition experienced a net force from the high concentration region to the low concentration region (Supplementary Fig. 30a). In previous studies, this unbalanced osmotic condition drove the motion of cancer cells and GUVs to the region with low osmolyte concentration following the osmotic engine model[31,32]. In our case, the forces generated from the unbalanced osmotic condition compressed the GUVs colony. This compression was better presented for the colony that

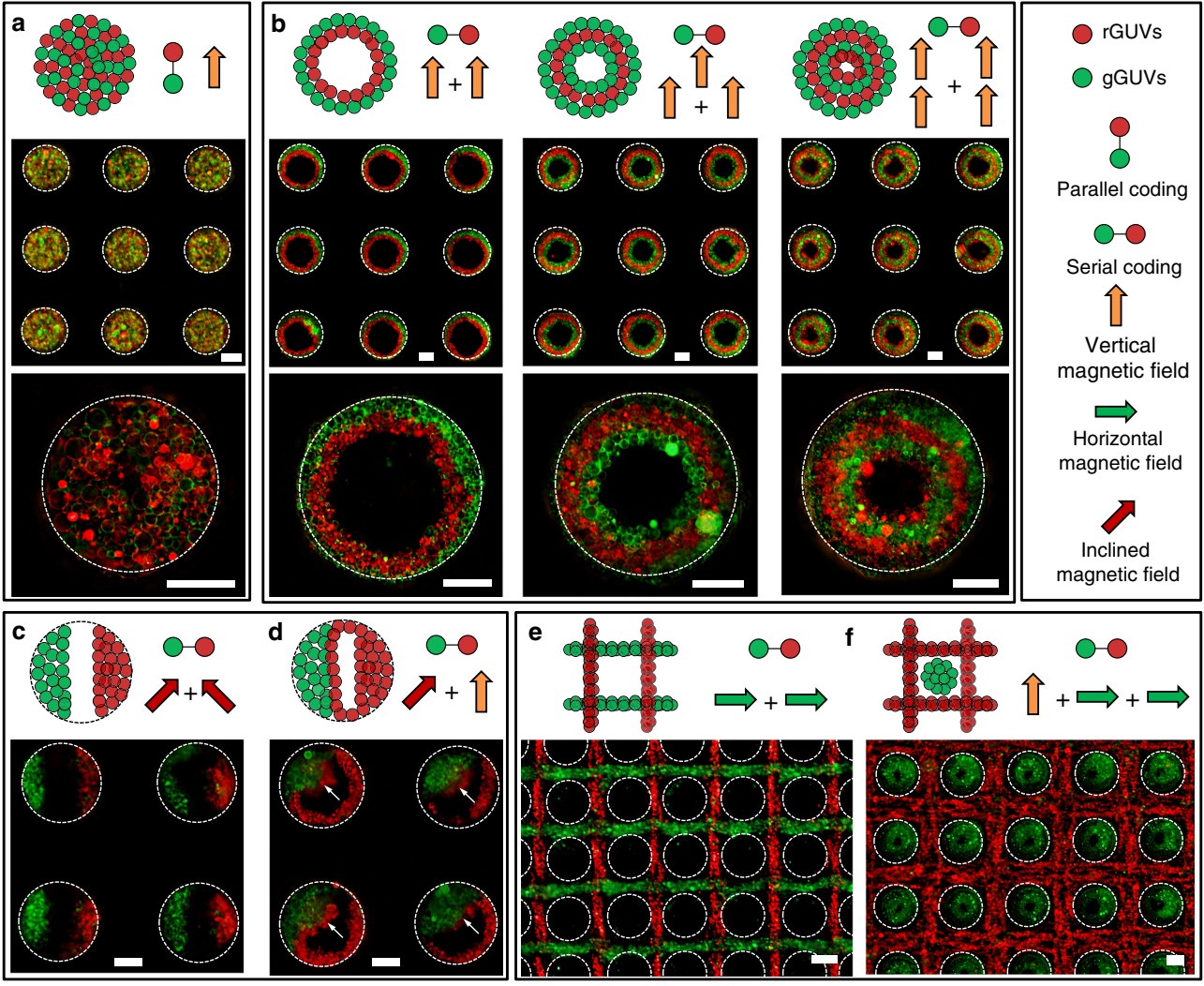

**Fig. 3 Coding of spatially controlled GUVs colonies. a** Schematic and fluorescence images of GUVs colonies formed via the parallel coding of giant unilamellar vesicles with green fluorescence (gGUVs) and giant unilamellar vesicles with red fluorescence (rGUVs). Schematic and fluorescence images of the serially coded GUVs colonies via application of vertical magnetic field for the alternative assembly of gGUVs and rGUVs (**b**), successive application of two inclined magnetic fields with different directions respectively for gGUVs assembly (putting the SS mesh on one side of the magnet) and rGUVs (putting the SS mesh on the other side of the magnet) (**c**), successive application of inclined magnetic field for gGUVs assembly and vertical magnetic field for rGUVs assembly (**d**), successive application of two perpendicular horizontal magnetic fields for the chronological assembly of gGUVs and rGUVs (**e**), and successive application of vertical magnetic field for gGUVs assembly and two perpendicular horizontal magnetic fields for rGUVs assembly (**f**). The dash circles indicated the microwell wall. The scale bars are 100 μm.

partially occupied the microwell. Under isotonic condition, the newly assembly GUVs colony contained some protrusions (yellow dash line in the left image of Supplementary Fig. 31). When external solution was replaced with 1 M glucose solution to introduce a hypertonic osmotic stress, the GUVs colony was compressed and the osmotic stress smoothed the interface between the GUVs colony and external solution (middle image in Supplementary Fig. 31). Protrusions reappeared when the hypertonic external solution was replaced with hypotonic pure water (right image in Supplementary Fig. 31). The compression from the external hypertonic osmotic stress sealed the voids among GUVs near the external solution, resulting in the failure of osmolyte and fluorescent dyes to penetrate into the GUVs colony (Supplementary Fig. 30b) as confirmed by Supplementary Fig. 30c–f. In isosmotic assembly solution containing $MnCl_2$, resorufin molecules in external solution gradually diffused into

the voids of GUVs colonies, as evidenced by the gradually enhanced red fluorescence intensity of resorufin with time in Supplementary Fig. 30c. In hypertonic glucose, NaCl, or $CaCl_2$ solutions, the red fluorescence of resorufin was not observed (Supplementary Fig. 30d–f), indicating its failure to penetrate into the GUVs colony. The block of the GUVs colony to small molecules made it behave as an elastic collective under hypertonic osmotic stress. The osmotic compression increased the elasticity energy of the GUVs colony, resulting in a negative hydrostatic energy in the GUVs colony that promoted the entering of water into the colony. The negative hydrostatic energy (promoting water in and elasticity energy release) balanced the osmotic stress (promoting water out) to stabilize the GUVs colony under hypertonic conditions. This osmotic stability of GUVs colonies endowed them as robust models for widespread applications. Moreover, it may also provide a tentative clue for the formation

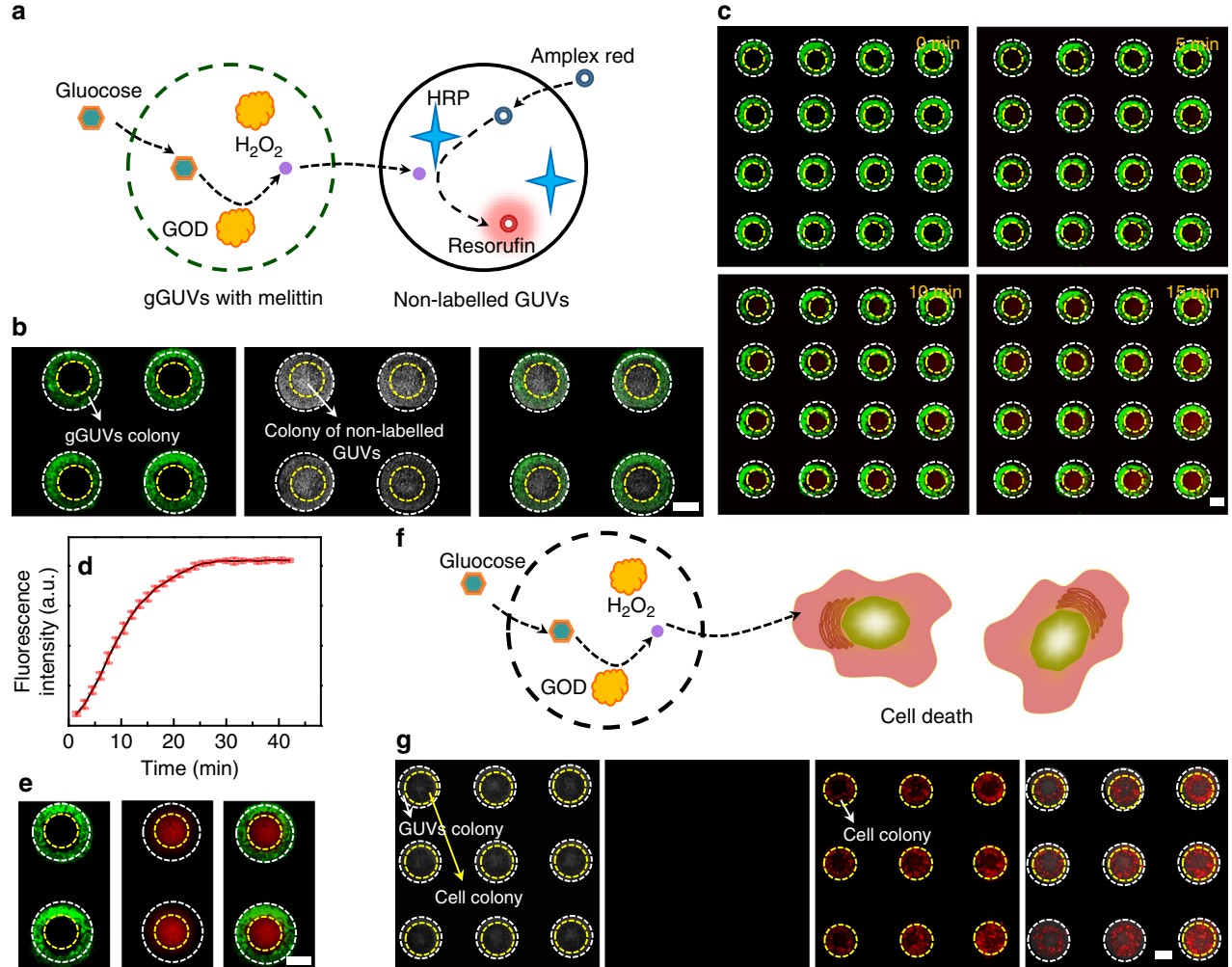

**Fig. 4 Spatialized biochemical reactions in tissue-like GUVs aggregates. a** Schematic illustration for the chemical communication between the colony of gGUVs with melittin and the non-labeled GUVs colony. **b** Fluorescence image and bright field image of the GUVs aggregates with two coaxial GUVs colonies: fluorescence image of the colonies of gGUVs with melittin (left), bright field image of the two kind of colonies (middle), merged image (right). **c** Fluorescence images of the GUVs aggregates against time after the biochemical reaction was initiated by the addition of glucose and Amplex Red. **d** Variation of the fluorescence intensity of the product against time. The error bar represents the standard error of mean (SEM), $n = 3$ independent experiments. **e** Fluorescence images of the tissue-like GUVs aggregates with fluorescent resorufin product: fluorescent image of gGUVs colony (left), fluorescent image of the product of resorufin (middle), merged image (right). **f** Schematic illustration of the cell death caused by $H_2O_2$ that is generated by the GUVs colonies. **g** Images for the $H_2O_2$ caused cell death: bright field image of GUVs colonies and cell colonies (the first one), fluorescence image of live cells (the second one), fluorescence image of dead cells (the third one), and merged image (the last one). GOD in **a** and **f** represents glucose oxidase. HRP in **a** represents horseradish peroxidase. The scale bars were 100 μm. Source data are provided as a Source Data file.

of proto-tissues from protocells on the early earth, since the membrane-based protocells aggregates exhibit a mechanical stability in face of external shock.

**Spatialized cascade reaction in the colonies.** One unique property of cells is their ability to spatialize biochemical reactions among organelles or tissues for efficient biosynthesis or precise signaling. In the following part, we evaluated the ability of the spatially coded GUVs aggregates to mimic the spatialized biochemical process in a simplified model. Two kinds of GUVs, i.e., gGUVs with melittin pores encapsulated with glucose oxidase (GOD) and non-labeled GUVs encapsulated with horseradish peroxidase (HRP), were magnetically assembled into coaxially coded colonies in the microwells under vertical magnetic field (Fig. 4a, b). With the addition of glucose and Amplex Red in the external solution, glucose entered gGUVs through melittin pores, and the uncharged Amplex Red passively diffused across the lipid

bilayers[33]. The GOD in gGUVs catalyzed the oxidation of glucose to generate $H_2O_2$, which diffused into the non-labeled GUVs, where they reacted with Amplex Red under the catalysis of HRP to generate the product of resorufin with red fluorescence (Fig. 4a). The non-labeled GUVs contained no protein pores, and $H_2O_2$ would also not oxidize the membranes to generate membrane defects[34,35] (Supplementary Fig. 32), so the charged resorufin molecules were trapped in the GUVs. With the increase of time, more resorufin molecules were formed, so the fluorescence intensity increased, until a plateau was achieved above 30 min (Fig. 4c, d). The red fluorescence of resorufin was mainly observed in the non-labeled GUVs (Fig. 4e), which indicated the ability of the magnetically coded GUVs aggregates to compartmentalize and spatialize biochemical reactions mimicking natural tissues. Except for the small molecules mediated chemical communication between artificial tissue-like assemblies, this technique can also be utilized to study the chemical process between GUVs colonies and cell colonies (Fig. 4f, g, and Supplementary

Fig. 33). The GUVs encapsulated with GOD generated $H_2O_2$, which diffused to the cell colony and caused cell death. The live cells were stained with fluorescein diacetate (FDA) with green fluorescence, and dead cells were labeled by propidium iodide (PI) with red fluorescence. After 6 h of incubation in 400 mM glucose solution, almost all the cells died as indicated by the negligible green fluorescence in the second image of Fig. 4g and the evident red fluorescence in the third image of Fig. 4g. As a control, GUVs colonies contained no cell death because of the absence of red fluorescence (Supplementary Fig. 34). According to the above experimental result, this technique holds great potential in the investigation of more complicated biological processes for the study of cell biology and the development of tissue models with higher-order collective behaviors.

In summary, we obtained tissue-mimicking GUVs aggregates arrays with different morphologies and spatially coded configurations using a SS mesh under magnetic field. GUVs in these aggregates exhibited uncustomary stability in hypotonic or hypertonic conditions in comparison with individual GUVs suspensions, which made them robust models for application in synthetic biology and cell biology, and suggests possible clues for the evolution of multicellular cells on early earth. Via the spatial coding of GUVs or cells, designated GUVs were illumined and cell death was triggered by enzyme reactions, proving the ability of the model to mimic the spatialized biochemical processes in natural tissues. This work paved the way for the study of higher-order tissue behaviors via the groundbreaking manipulation of diamagnetic objects into defined 3D structures.

## Methods

**Materials**. 1,2-dimyristoyl-*sn*-glycero-3-phosphocholine (DMPC), 1,2-dipalmi-toyl-sn-glycero-3-phosphocholine (DPPC), and 1,2-dioleoyl-*sn*-glycero-3-phos-pho-L-serine (sodium salt) (DOPS) were purchased from Avanti Polar Lipids (USA). 1,2-dioleoyl-sn-glycero-3-phosphoethanolamine-N-(7-nitro-2-1,3-benzox-adiazol-4-yl) (NBD PE) and Texas red labeled 1,2-dihexadecanoyl-*sn*-glycero-3-phosphoethanolamine, triethylammonium salt (TR-DHPE) were obtained from Invitrogen (China). Sucrose, glucose, HRP, GOD, Amplex Red, melittin, FDA, PI, manganese (II) chloride, and Gadobutrol were purchased from Sigma (China). Cylindrical NdFeb magnets (1T, diameter = 3 cm, thickness = 1 cm) were bought from Gates Qiangci Company (Shanghai, China). The square SS mesh (1 cm × 1 cm) with thickness of 100 μm was custom-made by RGRS Company (Shenzhen, China). Indium tin oxide (ITO) electrode was purchased from Hangzhou Yuhong technology Co. Ltd (China). Millipore Milli-Q water with a resistivity of 18.2 MΩ cm was used in the experiments.

**Giant unilamellar vesicles (GUVs) formation**. Two kinds of mother GUVs samples, i.e., DMPC/NBD PE (*w/w*, 95/5) GUVs and DMPC/DOPS/TR-DHPE (*w/w/w*, 95/4.5/0.5) GUVs, were formed using electroformation method in 400 mM sucrose solution using two face-to-face electrode layout of ITO electrodes. Lipid thin films were formed in the following procedure: 20 μL of lipid solution (5.0 mg mL$^{-1}$) was deposited on ITO electrode, and spread using a needle, followed by drying in a vacuum desiccator for 2 h. The two slides of ITO electrodes were then assembled with a 2 mm thick Teflon spacer with a 2 cm × 1 cm hole, as reported elsewhere[28,36,37]. To form GUVs, the electroformation instrument was placed on a hot plate with temperature of 45 °C, and an AC electric field with amplitude of 5 V and frequency of 10 Hz was applied for 2 h. The GUVs were observed under the fluorescence microscope.

**GUVs colonies formation**. GUVs colonies were formed under magnetic field in a home-made device (Supplementary Fig. 5). The device was assembled by adhering a square Teflon cell with opening size of ~1.1 cm × 1.1 cm to a cover slip using vacuum grease, followed by putting the SS mesh on the top of the cover slip in the cell. Before using it for GUVs assembly, the device was firstly treated in below procedure to avoid GUVs rupture during GUVs entrapment experiments. A total of 200 μL of DPPC ethanol-water solution with ethanol volume percentage of 40% and DPPC concentration of 0.10 mg mL$^{-1}$ (similar composition used for bicelles formation[38-40] by us) were added in the cell. The device was then heated at 50 °C for 5 min, and washed using 133 mM MnCl$_2$ solution for at least three times, resulting in the formation of supported DPPC membranes on the cover slip and SS mesh. To form GUVs colony arrays, GUVs mother dispersion in 400 mM sucrose solution was mixed with isotonic MnCl$_2$ (133 mM) solution to obtain a mixture with volume of 300 μL. The mixture was added in the cell, and then the device was put in magnetic field generated by NdFeB magnets. The GUVs concentration was

controlled by varying the volume ratio of GUVs sucrose solution and 133 mM MnCl$_2$ solution with fixed final solution volume of 300 μL. Magnetic fields with three different directions were used in the GUVs colonies formation experiments. The vertical magnetic field was provided by putting the SS mesh on the top center of the magnet. The horizontal magnetic field was provided by putting the SS mesh between two face-to-face magnets. An inclined magnetic field was provided by putting the SS mesh on one side of the top of the magnet. To obtain colonies with coded GUVs assemblies, GUVs solution containing different GUVs were successively added, or magnetic fields with different directions were successively applied. The most GUVs magnetic entrapment experiments were lasting more than 2 h.

**Spatialized chemical communications**. The communication between two GUVs populations or one GUVs population and one cell colony was investigated. For the study of communication between two GUVs populations, gGUVs and non-labeled GUVs were electroformed in 400 mM sucrose solution containing 12 μg mL$^{-1}$ GOD and 1.2 μg mL$^{-1}$ HRP, respectively. gGUVs encapsulated with GOD was firstly magnetically trapped in the microwells under vertical magnetic field, incu-bated in solution with 12 μg mL$^{-1}$ melittin for 2 h, and carefully washed with excess MnCl$_2$ solution to remove the non-encapsulated GOD and free melittin. Then the non-labeled GUVs encapsulated with HRP were added and trapped in the microwells under vertical magnetic field followed by the remove of non-encapsulated HRP via careful washing with MnCl$_2$ solution. To initiate the reac-tions, the external solution was replaced with 400 mM glucose containing 50 μM Amplex Red. The fluorescent product of the cascade reaction was monitored using fluorescence microscope. For the investigation of the communication between GUVs population and cell colony, non-labeled GUVs encapsulated with GOD were trapped in microwells, incorporated with melittin, and washed with excess Gadobutrol solution. Then HEPG2 cells were added to form coaxial aggregates of GUVs population (encapsulated with GOD) and cell colony. For comparison, coaxial aggregates of GUVs population with no GOD and cell colony were also fabricated. The two microstructures were then incubated in 400 mM glucose solution for 6 h. The communication between GUVs and cells was verified via the check of cell viability. Live and dead cells were stained with FDA and PI, respectively.

**Characterization**. The topology of the SS mesh was characterized by fluorescence microscope (Olympus IX73, Japan) and scanning electron microscopy (Quanta 200 FEG, Netherlands). The fluorescence images of the GUVs colonies were obtained by fluorescence microscope and laser confocal microscope (Olympus FV 3000, Japan).

**Simulation**. The magnetic field distribution around the SS mesh was simulated using COMSOL Multiphysics 4.3 software. The magnetic susceptibilities of GUVs, SS mesh, and 133 mM MnCl$_2$ solution were $-1.0 \times 10^{-5}$, 2, and 0.03, respectively. The external magnetic field generated by magnets magnetized the SS mesh for GUVs assembly.

**Reporting summary**. Further information on research design is available in the Nature Research Reporting Summary linked to this article.

## Data availability
The source data underlying Fig. 4d and Supplementary Figs. 6b, 9a, b, c, 11b, 32b, and d are provided as a Source Data file. The data that support the findings of this study are available within the paper and its supplementary information. All other relevant data are available from the authors upon reasonable request.

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

## Acknowledgements

This work was supported by the National Natural Science Foundation of China (Grant No. 21773050), the Natural Science Foundation of Heilongjiang Province for Distinguished Young Scholars (JC2018003).

## Author contributions

X.J.H. supervised the research. X.J.H., Q.C.L., and S.B.L. conceived and designed the experiments. Q.C.L., S. B. L., X. X. Z., and W. L. X. performed experiments. X.J.H., Q.C.L., S. B. L., X. X. Z., and W. L. X. analyzed the data. X.J.H., Q.C.L., and S. B. L wrote the paper, and all authors commented on the paper. Q. C. L and S. B. L contributed equally to this work.

## Competing interests

The authors declare no competing interests.
