## [Peer Review File · Nature Communications]

Reviewers' Comments:

Reviewer #1:

Remarks to the Author:

This manuscript describes experimental observations of the magnetic field-directed assembly of giant lipid vesicles into three-dimensional aggregates. The authors show that the close-packed vesicle aggregates adopt morphologies and patterns prescribed by (1) the direction of the field gradient via magneto-archimedes effect; (2) the concentrations of GUVs; (3) shapes and sizes of the container microwells; and (4) the order in which materially different GUVs are introduced. The authors report that the aggregates acquire an unusual insensitivity to osmotic stresses from the environment and illustrate how a multi-compartmental architecture of the aggregates can spatially pattern cascade chemical reactions. The latter demonstrated using glucose oxidase mediated production of hydrogen peroxide in one type of GUVs, whose diffusion into the second distinct type of compartments drives HRP mediated conversion of Amplex Red in one case and apoptotic killing of live cells in the other.

The experimental design is simple and innovative, observations insightful, and findings appear to support the central claim of the predictive organization of GUVs into tissue-like multicompartmental morphologies.

The work, to the best of my knowledge and understanding, represents a significant advance in organizing populations and colonies of vesicular compartments into mesoscale tissue-like structures with well-defined spatial patterns. Such constructs should prove useful for fundamental studies of physical-chemical consequences of multicellularity and might enable the design of tissue-mimetic models and materials. Given the broad appeal and the significance of the advance, I am inclined to recommend the publication of the work in Nature Communication. But, as presented, the manuscript begs a number of not-so-minor technical questions and lacks mechanistic insights, addressing which in my view is critically important.

1. As described, it appears that GUV containing 400 mM sucrose experience 400 mM sucrose and 133 mM MnCl_2 in the surrounding bath. This would create a 399 mM osmotic imbalance unless sucrose from the exterior bath is removed. A clearer explanation of the experimental conditions is needed. For the rest of the questions, I assume that the authors have 400 mM sucrose in the vesicular interior and 133 mM paramagnetic bath devoid of any sucrose.

2. Assuming that the GUVs experience osmotically balanced isotonic conditions, their bounding membranes must exhibit thermally excited out-of-plane fluctuations, which are determined by the bending modulus of the DMPC bilayer. One consequence of these so-called Helfrich undulations of the membrane surface is then a net repulsion between the membranes of individual GUVs arising from their entropic confinement, which is rather long-ranged. In addition, there are other headgroup and hydration mediated entropic repulsion that push the membranes apart. In light of these considerations, the proposed close-packing of GUVs into a consolidated aggregate then begs a number of questions which remain unanswered: (a) How do single GUVs deform in the presence of magnetic field? (B) How is the repulsion overcome to achieve closed packing? (c) Is the packed assembly fluid allowing individual GUVs to exhibit transitional mobility (d) Does the assembly remain stable upon removal of the applied field? If yes, what forces drives this irreversible transition?

3. How does the size polydispersity pattern the organization of GUVs within the aggregates? GUVs formed by the electroformation process naturally exhibits a rather broad variation in sizes. Since the areas of adhesive contact varies during "close packing" with size, one may expect size patterning within the consolidated aggregates.

4. What is the origin of the osmotic stability of the clusters of GUVs? Little or no mechanistic insight is offered on why the aggregation of vesicles renders them insensitive to osmotic stresses

from the environment. This is plainly counter-intuitive and begs a clear explanation and/or a testable hypothesis.

5. A clear evidence is needed to establish that the individual GUVs within the aggregates do not fuse, hemi-fuse, and retain the integrity of their compartmentalized interior.

6. It is not clear how Amplex Red enter the non-labelled GUVs. Why doesn't hydrogen peroxide oxidize lipids and permeabilize GUVs? Does the induced cell-death detach cells from the aggregates?

Minor: the use of the verb swarming appears misleading. There is little or no evidence of swarming dynamics during the assembly of the aggregate.

Reviewer #2:

Remarks to the Author:

The paper of Li et al. describes magnetically manipulated assembly of random giant unilamellar vesicles (GUVs) into spatially controlled patterns and can even do that sequentially adding other GUVs or else and add an extra level of complexity to the final assemblies. They perform such assemblies via using a stainless steel mesh and a magnetically responsive paramagnetic solution which enables to manipulate non-magnetic species. Authors claim that the assemblies obtained are 3D but indeed the structures have a very small aspect ratio and being also templated by a 2D mesh these obtained structures are not fully 3D in my view. There is limited additivity in the 3rd dimension of the constructions. Authors may want to improve the 3D aspect ratio. Here I mean the aspect ratio of the height of the assembly to the bottom dimension of the mesh pattern. Aspect ratio currently is 0.5 and I believe it is not enough to call it 3D. Showing additivity could be another aspect authors may consider in their revision.

The authors address similar capabilities as of the assemblies demonstrated in Ref 13. The capabilities and the demonstrated reactions/fusions etc. are shown in Fig.4 Here, they produce peroxide and diffuse it to neighbor cells to kill them or to catalyze another reaction monitored via fluorescence intensity in neighboring GUVs. These demonstrations support their claims of GUV assemblies being tissue-mimics at the introduction of the paper.

Authors mention in line 63 that the magnetic manipulation should be revolutionized to obtain further complex assemblies. I do think they have built their technique based on previously demonstrated assembly schemes (I donot see a revolution) as they have promptly referred. They have enriched these schemes by sequential assembly and sequence of magnetic field direction. Thus they have demonstrated interesting behavior in assembly but also in the spatial distribution of GUVs etc. The paper exhibit pieces of novelties in its content and can be an interesting read for the Nature Communications readers after revisions as mentioned.

For the sake of clarity, the comments of the reviewer have been collated in black and *italic*, and our response to each comment appears in blue. All the changes to the text of the manuscript are highlighted in red.

Reviewers' comments:

Reviewer #1 (Remarks to the Author):

This manuscript describes experimental observations of the magnetic field-directed assembly of giant lipid vesicles into three-dimensional aggregates. The authors show that the close-packed vesicle aggregates adopt morphologies and patterns prescribed by (1) the direction of the field gradient via magneto-archimedes effect; (2) the concentrations of GUVs; (3) shapes and sizes of the container microwells; and (4) the order in which materially different GUVs are introduced. The authors report that the aggregates acquire an unusual insensitivity to osmotic stresses from the environment and illustrate how a multi-compartmental architecture of the aggregates can spatially pattern cascade chemical reactions. The latter demonstrated using glucose oxidase mediated production of hydrogen peroxide in one type of GUVs, whose diffusion into the second distinct type of compartments drives HRP mediated conversion of Amplex Red in one case and apoptotic killing of live cells in the other.

The experimental design is simple and innovative, observations insightful, and findings appear to support the central claim of the predictive organization of GUVs into tissue-like multicompartmental morphologies.

The work, to the best of my knowledge and understanding, represents a significant advance in organizing populations and colonies of vesicular compartments into mesoscale tissue-like structures with well-defined spatial patterns. Such constructs should prove useful for fundamental studies of physical-chemical consequences of multicellularity and might enable the design of tissue-mimetic models and materials. Given the broad appeal and the significance of the advance, I am inclined to recommend the publication of the work in Nature Communication. But, as presented, the manuscript begs a number of not-so-minor technical questions and lacks mechanistic insights, addressing which in my view is critically important.

1. As described, it appears that GUV containing 400 mM sucrose experience 400 mM sucrose and 133 mM MnCl₂ in the surrounding bath. This would create a 399 mM osmotic imbalance unless sucrose from the exterior bath is removed. A clearer explanation of the experimental conditions is needed. For the rest of the questions, I assume that the authors have 400 mM sucrose in the vesicular interior and 133 mM paramagnetic bath devoid of any sucrose.

Thank the reviewer for the comment. The assembly solution was formed via mixing of GUVs mother dispersion in 400 mM sucrose solution with isotonic (133 mM) MnCl_2 solution. The volume ratio for most cases in this paper was 1:4. Therefore, in the assembly solution, the inner volume of GUVs was occupied by 400 mM sucrose solution, and the outside solution was an isotonic mixture of sucrose and MnCl_2 . We made below changes in the manuscript.

Page 4, “The assembly of diamagnetic GUVs was carried out on a stainless steel (SS) mesh (thickness $\approx 100 \mu\text{m}$) with patterned microwells (Figure S1) by mixing GUVs mother dispersion electroformed in 400 mM sucrose solution (Figure S2) with isotonic paramagnetic MnCl_2 or Gadobutrol solution. The inner volume of GUVs was 400 mM sucrose solution, and the outside solution was an isotonic mixture of sucrose and paramagnetic compounds with relative lower density.”

Page 6, “Complete occupation of the microwells to form columnar GUVs colonies arrays (Figure 1D) was realized with incubation time of 2h in 0.04 mg/mL GUVs solution (volume = 300 μL , $V_{\text{GUVs mother solution}}/V_{\text{MnCl}_2 \text{ solution}}=1/4$).”

Page 21, “To form GUVs colony arrays, GUVs mother dispersion in 400 mM sucrose solution was mixed with isotonic MnCl_2 (133 mM) solution to obtain a mixture with volume of 300 μL . The mixture was added in the cell, and then the device was put in magnetic field generated by NdFeB magnets. ”

2. Assuming that the GUVs experience osmotically balanced isotonic conditions, their bounding membranes must exhibit thermally excited out-of-plane fluctuations, which are determined by the bending modulus of the DMPC bilayer. One consequence of these so-called Helfrich undulations of the membrane surface is then a net repulsion between the membranes of individual GUVs arising from their entropic confinement, which is rather long-ranged. In addition, there are other headgroup and hydration mediated entropic repulsion that push the membranes apart. In light of these considerations, the proposed close-packing of GUVs into a consolidated aggregate then begs a number of questions which remain unanswered:

(a) How do single GUVs deform in the presence of magnetic field?

Thank the reviewer for the question. We observed no obvious deformation of single GUVs under magnetic field as shown in Figure S8.

We added below sentence in page 6 of the manuscript.

“The gravity facilitated the deposition of GUVs around the mesh and the magnetic force promoted their aggregation at region with lower magnetic field rather than deformed them (Figure S8).”

Figure S8. Fluorescence images of a single GUV with no application of magnetic field (left) and under ~ 0.2 T vertical magnetic field (right). The scale bar was $20\ \mu\text{m}$.

(B) How is the repulsion overcome to achieve closed packing?

Thank the reviewer for the question. In this work, the close packing of GUVs in the colony was achieved via the assembly of GUVs from the gravity of GUVs and the magnetic force exerted on GUVs. The interaction forces among GUVs had negligible effect on the close packing.

As indicated by the reviewer, there existed different kinds of repulsive forces among GUVs, including undulation, hydration, and electrostatic forces. The electrostatic repulsive force originated from negatively charged fluorescent lipid molecules (NBD PE), and the zwitterionic DMPC lipids due to the orientation of the headgroups and hydration layers¹. The assembly mixture was formed via the mixing of 133 mM MnCl_2 solution and 400 mM sucrose solution containing GUVs, resulting a final mixture with MnCl_2 concentration of ~ 106 mM. Electrostatic repulsion was screened in the salt-containing solution, allowing the attractive Van der Waals force to dominate for GUVs adhesion. The adhesion was verified by the existence of GUVs aggregates after the colony was mechanically separated from the microwell (right image in Figure S7). Ces and Elani observed similar adhesion of GUVs in 0.75 M NaCl solution¹.

However, although the forces among GUVs were involved in GUVs adhesion, they had negligible effects on the close packing of GUVs in the microwells. For GUVs colony formed in nonionic Gadobutrol solution (Figure S7), the GUVs were repulsive with each other as the nonionic Gadobutrol failed to screen the electrostatic repulsive force, so the GUVs colony disassembled into free and separate GUVs when the GUVs colony was mechanically separated from the microwell. Nevertheless, as shown by the fluorescence images in Figure S7C, the repulsive GUVs in the colony formed in Gadobutrol solution were also closely packed, and displayed mechanical stability as a collective when the mesh was inverted, similar to the case in the assembly solution containing MnCl_2 .

The GUVs encapsulated with sucrose solution were heavier than their surroundings. During the assembly process, the gravity facilitated the deposition of GUVs around the mesh and the magnetic force promoted their assembly into the microwells. These two mechanical forces overcame the repulsive forces among GUVs and induced close

packing. The involvement of mechanical forces for close packing has been previously reported in other work. For instance, gravity promoted the close packing of repulsive emulsions^{2,3}.

We added below content in page 6 of the manuscript.

“There existed the adhesive van der Waals force and different kinds of repulsive forces, including undulation, hydration, and electrostatic forces, among GUVs. The balance of these forces determined whether the GUVs were repulsive or adhesive. However, this had no influence on the close packing of GUVs. The adhesive GUVs in assembly solution containing $MnCl_2$, and the repulsive GUVs in 400 mM nonionic Gadobutrol solution can both form closely packed GUVs colonies (Figure S7). The driving forces for GUVs assembly and close packing were the magnetic force and the gravity.”

Figure S7. Stability of the GUVs assembly. A, Fluorescence image of GUVs aggregates in the assembly solution containing $MnCl_2$ after removal of magnetic field (left), being inverted for 60 min with no magnetic field application (middle), and being mechanically disrupted (right). B, Fluorescence image of GUVs aggregates in 400 mM glucose solution after removal of magnetic field (left), being inverted for 60 min with no magnetic field application (middle), and being mechanically disrupted

(right). C, Fluorescence image of GUVs aggregates in 400 mM Gadobutrol solution after removal of magnetic field (left), being inverted for 60 min with no magnetic field application (middle), and being mechanically disrupted (right). The scale bars were 100 μm .

(c) Is the packed assembly fluid allowing individual GUVs to exhibit transitional mobility?

Thank the reviewer for the question. We monitored the vesicle distribution in a GUVs colony with time under the fluorescence microscope. As shown by Video S1 and the typical fluorescence images in Figure S10, only very small local oscillation was observed in the GUVs colony, indicating that the GUVs colony were in jammed state. Jamming made GUVs in the colony behave as a collective and inhibited the transitional mobility of individual GUVs.

We added below sentence in page 7 of the manuscript.

“However, these GUVs displayed no individual transitional motion (Video S1, Figure S10), but behaved as a jammed and consolidated aggregate with collective stability due to their close packing.”

Figure S10. Fluorescence images of one GUVs colony composed of GUVs with green fluorescence (labeled with NBD PE) and GUVs with red fluorescence (labeled with TR DHPE) at 0 min and 110 min. The scale bar was 20 μm .

(d) Does the assembly remain stable upon removal of the applied field?

Thank the reviewer for the question. The assembly remain stable upon removal of the applied field.

Two kinds of fields were involved in the assembly of dispersed GUVs around the mesh. The gravity field contributed to the deposition of GUVs encapsulated with sucrose around the mesh, and the magnetic field devoted to assembling these GUVs into tissue-like structures with different morphology configurations. The magnetic field can be removed by the taking away of the magnet, and the supportive gravity field can be reversed to a harmful one that promotes disassembly via inverting the mesh. In our experimental results, taking away of the magnetic and inverting the mesh caused no disassembly of the GUVs aggregates (Figure S7A), which indicated that the assembly was stable after the removal of fields that promoted assembly formation.

If yes, what forces drives this irreversible transition?

The stability of the GUVs colony was attributed to the close packing of GUVs in the microwells.

For assembly with closely packed particles, chain-like contact forces can be generated among particles^{4, 5, 6}, which make the assembly behave as a collective and stable under shearing or other external forces. In our case, when magnetic field was removed without inverting the mesh, the gravity of GUVs kept them in the microwell and the close packing hindered their motion; when magnetic field was removed and the mesh was inverted, the contact forces among the closely packed GUVs kept them as a collective and inhibited the separation of the GUVs colony from the microwell and its disassembly under the colony's own gravity.

We added below content in page 7 of the manuscript.

“When magnetic field was removed and the SS mesh was inverted to provide a harmful gravity field that promoted the GUVs colony disassembly, the contact forces²⁵⁻²⁷ among the closely packed GUVs kept the GUVs colony stable, regardless of whether the GUVs were mutually repulsive or adhesive (Figure S7).”.

Figure S7. Stability of the GUVs assembly. A, Fluorescence image of GUVs aggregates in the assembly solution containing MnCl_2 after removal of magnetic field (left), being inverted for 60 min with no magnetic field application (middle), and being mechanically disrupted (right). B, Fluorescence image of GUVs aggregates in 400 mM glucose solution after removal of magnetic field (left), being inverted for 60 min with no magnetic field application (middle), and being mechanically disrupted (right). C, Fluorescence image of GUVs aggregates in 400 mM Gadobutrol solution after removal of magnetic field (left), being inverted for 60 min with no magnetic field application (middle), and being mechanically disrupted (right). The scale bars were 100 μm .

3. *How does the size polydispersity pattern the organization of GUVs within the aggregates? GUVs formed by the electroformation process naturally exhibits a rather broad variation in sizes. Since the areas of adhesive contact varies during “close packing” with size, one may expect size patterning within the consolidated aggregates.*

We thank the reviewer for the question. The GUVs in our work were formed using electroformation method, and they were quite disperse as mentioned by the reviewer. As reported in previous work^{7,8}, the size polydispersity of hard particles can cause the distribution of different sized particles among different spaces to achieve random

close packing. The polydisperse GUVs in our work also tended to assemble into closely packed state. For instance, we observed the filling of the voids created among large GUVs by small GUVs as indicated by the yellow arrows in Figure S9.

However, in our case, the GUVs close packing was neither fully size dependent nor totally random. Firstly, GUVs were different from hard particles during the assembly process, they can adjust their shape to remove the voids; therefore large GUVs can also be closely packed via adhesion and deformation as indicated by the blue arrows in Figure S9. Secondly, during the magnetic assembly process, as larger GUVs settled more quickly than smaller ones under gravity⁹, they were trapped around the microwell walls firstly and smaller GUVs were trapped later. Therefore, the final GUVs assemblies displayed a gradual decrease of the GUVs diameter from microwell wall to the center (Figure S9A). This phenomenon was more pronounced for the GUVs colony that partially occupied the microwells via using fewer GUVs (Figure S9B). Based on this size-dependent assembly phenomenon, we programmed the formation of GUVs colony with alternating GUVs layers of different average sizes (Figure S9C) via the successive adding of GUVs for two times.

Taken together, the size polydispersity partially contributed to GUVs close packing and patterned the organization of GUVs aggregates with gradient size distribution.

We added below content in Page 6 of the manuscript.

“As shown by Figure 1E and Figure S9A, the electroformed polydisperse GUVs displayed heterogeneous size distribution in the microwells. From the area close to the microwell wall to that close to the center, the average diameter of GUVs gradually decreased. This phenomenon was more pronounced for the GUVs colony that partially occupied the microwells via using fewer GUVs (Figure S9B). This is because larger GUVs settled more quickly under gravity than smaller ones²⁴. Based on this size-dependent assembly phenomenon, we programmed the formation of GUVs colony with alternating GUVs layers of different average sizes (Figure S9C) via the successive adding of GUVs for two times.”

Figure S9. Size distribution of GUVs in the microwells. A, Fluorescence image of a GUVs colony that fully occupied the microwell (top) and the diagram for the variation of GUVs average size with different regions in the fluorescence image (bottom). B, Fluorescence image of a GUVs colony that partially occupied the microwell (top) and the diagram for the variation of GUVs average size with different regions in the fluorescence image (bottom). C, Fluorescence image of a GUVs colony via the successive addition of GUVs for two times (top) and the diagram or the variation of GUVs average size with different regions in the fluorescence image (bottom). The scale bars were 50 μm .

4. What is the origin of the osmotic stability of the clusters of GUVs? Little or no mechanistic insight is offered on why the aggregation of vesicles renders them insensitive to osmotic stresses from the environment. This is plainly counter-intuitive and begs a clear explanation and/or a testable hypothesis.

Thank the reviewer for the question. The osmotic stability of the GUVs colony originated from the resistance of the GUVs in the colony as a collective to the external osmotic shock.

GUVs in the colony were closely packed. In hypotonic conditions, for instance, pure water, the reinforced mechanical stability of individual GUVs from the crowded GUVs surroundings impeded their rupture and maintained their morphologies.

In hypotonic conditions, GUVs in unbalanced osmotic condition experienced a net force from the high concentration region to the low concentration region (Figure S30A). In previous studies, this unbalanced osmotic condition drove the motion of cancer cells and GUVs to the region with low osmolyte concentration following the “osmotic engine model”^{10, 11}. In our case, the forces generated from the unbalanced osmotic condition compressed the GUVs colony. This compression was better presented for the colony that partially occupied the microwell. Under isotonic condition, the newly assembly GUVs colony contained some protrusions (yellow dash line in the left image of Figure S31). When the external solution was replaced with 1 M glucose solution to introduce a hypertonic osmotic stress, the GUVs colony was compressed and the osmotic stress smoothed the interface between the GUVs colony and external solution (middle image in Figure S31). When the external solution was replaced with pure water, protrusions reappeared (right image in Figure S31).

The compression from the external hypertonic osmotic stress sealed the voids among GUVs near the external solution, resulting in the failure of osmolyte and fluorescent dyes to penetrate into the GUVs colony (Figure S30B) as confirmed by Figure S30C-F. In isosmotic assembly solution containing MnCl_2 , resorufin molecules in external solution gradually diffused into the voids of GUVs colonies, as evidenced by the gradually enhanced red fluorescence of resorufin with time in Figure S30C. In hypertonic glucose, NaCl , or CaCl_2 solutions, the red fluorescence of resorufin was not observed (Figure S30D-F), indicating its failure to penetrate into the GUVs

colony. The block of the GUVs colony to small molecules made it behave as an elastic collective under hypertonic osmotic stress. The osmotic compression increased the elasticity energy of the GUVs colony, resulting in a negative hydrostatic energy in the GUVs colony that promoted the entering of water into the colony. The negative hydrostatic energy (promoting water in and elasticity energy release) balanced the osmotic stress (promoting water out) to stabilize the GUVs colony under hypertonic conditions.

We added Figure S30 and S31, and below contents in page 14 and 15 of the manuscript.

“The osmotic stability of the GUVs colony originated from the resistance of the closely packed GUVs in the colony as a collective to the external osmotic shock. In hypotonic condition provided by pure water, the reinforced mechanical stability of individual GUVs from the crowded GUVs surroundings impeded their rupture and maintained their morphologies. In hypotonic conditions, GUVs in unbalanced osmotic condition experienced a net force from the high concentration region to the low concentration region (Figure S30A). In previous studies, this unbalanced osmotic condition drove the motion of cancer cells and GUVs to the region with low osmolyte concentration following the “osmotic engine model”^{31,32}. In our case, the forces generated from the unbalanced osmotic condition compressed the GUVs colony. This compression was better presented for the colony that partially occupied the microwell. Under isotonic condition, the newly assembly GUVs colony contained some protrusions (yellow dash line in the left image of Figure S31). When external solution was replaced with 1 M glucose solution to introduce a hypertonic osmotic stress, the GUVs colony was compressed and the osmotic stress smoothed the interface between the GUVs colony and external solution (middle image in Figure S31). Protrusions reappeared when the hypertonic external solution was replaced with hypotonic pure water (right image in Figure S31). The compression from the external hypertonic osmotic stress sealed the voids among GUVs near the external solution, resulting in the failure of osmolyte and fluorescent dyes to penetrate into the GUVs colony (Figure S30B) as confirmed by Figure S30C-F. In isosmotic assembly solution containing $MnCl_2$, resorufin molecules in external solution gradually diffused into the voids of GUVs colonies, as evidenced by the gradually enhanced red fluorescence of resorufin with time in Figure S30C. In hypertonic glucose, NaCl, or $CaCl_2$ solutions, the red fluorescence of resorufin was not observed (Figure D30D-F), indicating its failure to penetrate into the GUVs colony. The block of the GUVs colony to small molecules made it behave as an elastic collective under hypertonic osmotic stress. The osmotic compression increased the elasticity energy of the GUVs colony, resulting in a negative hydrostatic energy in the GUVs colony that promoted the entering of water into the colony. The negative hydrostatic energy (promoting water in and elasticity

energy release) balanced the osmotic stress (promoting water out) to stabilize the GUVs colony under hypertonic conditions.”

Figure S30. Mechanism of the GUVs colony’s osmotic stability. A, Schematic of GU in isosmotic condition and unbalanced hypertonic condition. B, Schematic of GUVs colony in isosmotic condition and hypertonic condition. C, Fluorescence images illustrating the diffusion of resorufin (red fluorescence) into the voids of GUVs colony (green fluorescence) in isosmotic assembly solution. D-F, Fluorescence images illustrating the failure of resorufin to enter in the GUVs colony voids in hypertonic 1 M glucose solution (D), 500 mM NaCl solution (E), and 333.3 mM CaCl₂ solution. The scale bars were 100 μm.

Figure S31. Fluorescence images of GUVs colony that partially occupied the microwell successively in isosmotic assembly solution containing MnCl_2 (left), 1 M glucose solution (middle), and pure water (right). The scale bars were 100 μm .

5. A clear evidence is needed to establish that the individual GUVs within the aggregates do not fuse, hemi-fuse, and retain the integrity of their compartmentalized interior.

Thank the reviewer for the suggestion. We observed the variation of the GUVs colony composed of red fluorescent GUVs (labeled with TR DHPE) and green fluorescent GUVs with time (labeled with NBD PE). As shown by video S1 and Figure S10, no fusion/hemi-fusion events happened, because no exchange of fluorescence labeled phospholipid among red fluorescent GUVs and green fluorescent GUVs were observed. The integrity of the compartmentalized interior of GUVs was verified via the observation of GUVs colony containing some GUVs that were encapsulated with red fluorescent Rhodamine B isothiocyanate-Dextran. As shown in Figure S11, no leakage of Rhodamine B isothiocyanate-Dextran was observed, indicating that the GUVs in the colony had integrated interiors separated from external solution and other GUVs.

We added below sentences in page 7 of the manuscript.

“The GUVs in the colony were mutually isolated and retained the integrity of their compartmentalized interior. No fusion/hemi-fusion events and leakage of compartmentalized fluorescent molecules were observed in the experiments (Video S1, Figure S10 and S11).”

Figure S10. Fluorescence images of one GUVs colony composed of GUVs with green fluorescence (labeled with NBD PE) and GUVs with red fluorescence (labeled with TR DHPE) at 0 min and 110 min. The scale bar was 20 μm .

Figure S11. Integrity study of the compartmentalized interior of GUVs in the colony. A, Merged fluorescence image of a GUVs (labeled with green NBD PE) colony containing some GUVs encapsulated with Rhodamine B isothiocyanate-Dextran (molecular weight, 70 kDa, red fluorescence). B, Normalized fluorescence intensity against time of the three GUVs in A. The scale bar was 50 μm .

6. *It is not clear how Amplex Red enter the non-labelled GUVs.*

Thank the reviewer for the question. Amplex Red is an uncharged molecule that freely diffuses from the bulk aqueous medium across the lipid bilayer to enter into the aqueous interior of the GUVs¹². So they enter the non-labelled GUVs in a passive diffusion way.

We added below sentence in page 16 of the manuscript.

“With the addition of glucose and Amplex Red in the external solution, glucose entered gGUVs through melittin pores, and the uncharged Amplex Red passively diffused across the lipid bilayers³³.”

Why doesn't hydrogen peroxide oxidize lipids and permeabilize GUVs?

Thank the reviewer for the question. Hydrogen peroxide belongs to the non-radical group of reactive oxygen species (ROS), which has less oxidization ability compared to other ROS molecules¹³. So hydrogen peroxide has negligible effects to membranes if it is not converted to radical ROS. The interaction of hydrogen peroxide with membranes containing unsaturated moieties has been previously investigated by Liao and Nakao^{14, 15}. No obvious influences on membrane composition and permeability were observed in their experimental results.

In our work, the composition of the GUVs were DMPC (containing no unsaturated moieties) for non-labelled GUVs and DMPC/NBD-DOPE (molecular ratio=95/5) for gGUVs, where NBD-DOPE contained unsaturated C=C bonds. Although previous work confirmed the negligible effects of hydrogen peroxide on membranes, we carried out permeability experiments to further confirm the resistance of membranes to the oxidation effects of hydrogen peroxide in our work (Figure S32). DMPC GUVs and DMPC/DOPE (molecular ratio, 95/5) encapsulated with calcein were formed. When those GUVs were dispersed in solutions containing 0.1 mM, 1 mM, or 10 mM hydrogen peroxide, we observed no obvious decrease of the fluorescence intensity of calcein with time, which indicated no leakage of calcein. Therefore hydrogen peroxide has negligible influence on membrane permeability.

We added below sentence in page 16 of the manuscript.

“The non-labelled GUVs contained no protein pores, and H₂O₂ would also not oxidize the membranes to generate membrane defects^{34,35} (Figure S32), so the charged resorufin molecules were trapped in the GUVs.”

Figure S32. Influence of hydrogen peroxide on GUVs membrane permeability. A, Fluorescence images of DMPC GUVs containing calcein against incubation time in hydrogen peroxide solution with different concentrations. B, Variation of the normalized fluorescence intensity of calcein in DMPC GUVs against incubation time in hydrogen peroxide solution with different concentrations. C, Fluorescence images of DMPC/DOPE (Molecular ratio, 95/5) GUVs containing calcein against incubation time in hydrogen peroxide solution with different concentrations. D, Variation of the normalized fluorescence intensity of calcein in DMPC/DOPE (Molecular ratio, 95/5) GUVs against incubation time in hydrogen peroxide solution with different concentrations. The scale bars were 20 μm .

Does the induced cell-death detach cells from the aggregates?

Thank the reviewer for the question. We observed no detached dead cells from the aggregates as shown by the fluorescence image in Figure 4G.

Minor: the use of the verb swarming appears misleading. There is little or no evidence of swarming dynamics during the assembly of the aggregate.

Thank the reviewer for the suggestion. We deleted the verb swarming in the manuscript.

Reviewer #2 (Remarks to the Author):

The paper of Li et al. describes magnetically manipulated assembly of random giant unilamellar vesicles (GUVs) into spatially controlled patterns and can even do that sequentially adding other GUVs or else and add an extra level of complexity to the final assemblies. They perform such assemblies via using a stainless steel mesh and a

magnetically responsive paramagnetic solution which enables to manipulate non-magnetic species. Authors claim that the assemblies obtained are 3D but indeed the structures have a very small aspect ratio and being also templated by a 2D mesh these obtained structures are not fully 3D in my view. There is limited additivity in the 3rd dimension of the constructions. Authors may want to improve the 3D aspect ratio. Here I mean the aspect ratio of the height of the assembly to the bottom dimension of the mesh pattern. Aspect ratio currently is 0.5 and I believe it is not enough to call it 3D. Showing additivity could be another aspect authors may consider in their revision.

Thank the reviewer for the suggestion. We carried out GUVs assembly experiments on stainless steel mesh with larger thickness to improve the 3D aspect ratio. GUVs colonies with 3D aspect ratio of ~ 1 and 1.5 were obtained as shown in Figure S13. These experimental results confirmed the ability of our method to form local 3D assemblies in the microwells. We added below description in Page 7 of the manuscript.

“Columnar GUVs colonies with aspect ratios (height/diameter) of 0.5 (Figure 1D), 1.0, and 1.5 (Figure S13) can be obtained via the variation of the microwell aspect ratios.”

Figure S13. Fluorescence images of GUVs colonies with 3D aspect ratio of ~ 1 (A) and ~ 1.5 (B). The scale bars were 100 μm .

The authors address similar capabilities as of the assemblies demonstrated in Ref 13. The capabilities and the demonstrated reactions/fusions etc. are shown in Fig.4 Here, they produce peroxide and diffuse it to neighbor cells to kill them or to catalyze

another reaction monitored via fluorescence intensity in neighboring GUVs. These demonstrations support their claims of GUV assemblies being tissue-mimics at the introduction of the paper.

Thank the reviewer for the comment.

Authors mention in line 63 that the magnetic manipulation should be revolutionized to obtain further complex assemblies. I do think they have built their technique based on previously demonstrated assembly schemes (I donot see a revolution) as they have promptly referred. They have enriched these schemes by sequential assembly and sequence of magnetic field direction. Thus they have demonstrated interesting behavior in assembly but also in the spatial distribution of GUVs etc. The paper exhibit pieces of novelties in its content and can be an interesting read for the Nature Communications readers after revisions as mentioned.

Thank the reviewer for the comment and recognition of our work. We have changed the sentence in line 63 (old version) into “However, the application of this technique for the formation of more complicated 3D aggregates, for instance, the spatially coded tissue-like GUVs assemblies, has rarely been reported.”. (page 3 in revised version)

References

1. Bolognesi G, *et al.* Sculpting and fusing biomimetic vesicle networks using optical tweezers. *Nat. Commun.* **9**, 1882 (2018).
2. Jorjadze I, Pontani L-L, Brujic J. Microscopic Approach to the Nonlinear Elasticity of Compressed Emulsions. *Phys. Rev. Lett.* **110**, 048302 (2013).
3. Maestro A, Drenckhan W, Rio E, Höhler R. Liquid dispersions under gravity: volume fraction profile and osmotic pressure. *Soft Matter* **9**, 2531-2540 (2013).
4. Zhou J, Long S, Wang Q, Dinsmore AD. Measurement of Forces Inside a Three-Dimensional Pile of Frictionless Droplets. *Science* **312**, 1631-1633 (2006).
5. Corwin EI, Jaeger HM, Nagel SR. Structural signature of jamming in granular media. *Nature* **435**, 1075-1078 (2005).
6. Brujić J, Edwards SF, Grinev DV, Hopkinson I, Brujić D, Makse HA. 3D bulk measurements of the force distribution in a compressed emulsion system. *Faraday Discuss.* **123**, 207-220 (2003).
7. Clusel M, Corwin EI, Siemens AON, Brujić J. A ‘granocentric’ model for random packing of jammed emulsions. *Nature* **460**, 611-615 (2009).
8. Heckendorf D, Mutch KJ, Egelhaaf SU, Laurati M. Size-Dependent Localization in Polydisperse Colloidal Glasses. *Phys. Rev. Lett.* **119**, 048003 (2017).
9. Varnier A, *et al.* A Simple Method for the Reconstitution of Membrane Proteins into Giant Unilamellar Vesicles. *J Membrane Biol.* **233**, 85-92 (2010).
10. Stroka Kimberly M, *et al.* Water Permeation Drives Tumor Cell Migration in Confined Microenvironments. *Cell* **157**, 611-623 (2014).

11. Shoji K, Kawano R. Osmotic-engine-driven liposomes in microfluidic channels. *Lab Chip* **19**, 3472-3480 (2019).
12. Piwonski HM, Goomanovsky M, Bensimon D, Horovitz A, Haran G. Allosteric inhibition of individual enzyme molecules trapped in lipid vesicles. *P. Natl. Acad. Sci. USA* **109**, E1437-E1443 (2012).
13. Bienert GP, Schjoerring JK, Jahn TP. Membrane transport of hydrogen peroxide. *BBA-Biomembranes* **1758**, 994-1003 (2006).
14. Yoshimoto M, Miyazaki Y, Umemoto A, Walde P, Kuboi R, Nakao K. Phosphatidylcholine Vesicle-Mediated Decomposition of Hydrogen Peroxide. *Langmuir* **23**, 9416-9422 (2007).
15. Tai W-Y, *et al.* Interplay between Structure and Fluidity of Model Lipid Membranes under Oxidative Attack. *J. Phys. Chem. B* **114**, 15642-15649 (2010).

Reviewers' Comments:

Reviewer #1:

Remarks to the Author:

Upon careful consideration of the Authors's responses and revisions in response to the questions raised during the initial review, I feel the manuscript is publishable in Nature Communications.

I recommend the publication of the current version of the manuscript.

Reviewer #2:

Remarks to the Author:

I have read the documents and saw the detailed changes made on my comments but also on the changes made in response to the other referee. I believe the paper is now ready for publication in Nature Communication with many necessary information in the SI for curious readers.

Reviewer #1 (Remarks to the Author):

Upon careful consideration of the Authors's responses and revisions in response to the questions raised during the initial review, I feel the manuscript is publishable in Nature Communications.

I recommend the publication of the current version of the manuscript.

Thank the reviewer for the careful review and the recognition of our work.

Reviewer #2 (Remarks to the Author):

I have read the documents and saw the detailed changes made on my comments but also on the changes made in response to the other referee. I believe the paper is now ready for publication in Nature Communication with many necessary information in the SI for curious readers.

Thank the reviewer for the review, comment and recognition of our work.